# How social media influences the self-image and body image among female and male adolescents

**Merete Kolberg Tennfjord**[1*], **Ashley Rebecca Bell**[2], **Ragnhild Eg**[3]

**1** Department of Health and Training, School of Health Sciences, Kristiania University of Applied Sciences, Oslo, Norway, **2** Department of Psychology, Pedagogy and Law, School of Health Sciences, Kristiania University of Applied Sciences, Oslo, Norway, **3** Innovation, Consumer and Sensory Sciences, Nofima, Ås, Norway

* meretekolberg.tennfjord@kristiania.no

## Abstract

Parents, peers, and social media are the primary drivers of sociocultural influence among young people, where social media is considered the most pervasive and persuasive of these drivers. However, there is a need for a more nuanced understanding of this influence, considering the differences in how young males and females view their self-images and body images. This study combined qualitative data from focus-group interviews and a national survey to investigate social media influences on Norwegian adolescents' body image and self-image. Forty-eight adolescents (58% females) attending secondary schools mean age 15.9 (age range 15–19 years) participated in eight focus-group interviews. Their responses to the questions about social media's influence on body image were analyzed thematically. The focus group interviews were succeeded six months later by a national survey administered to adolescents (59% females) attending secondary schools with a mean age of 18 (age range 16–20 years). One-hundred-twenty-four responses to social media's impact on their self-images were analyzed both quantitatively and qualitatively. Triangulating the data suggested a tendency for social media to have a negative impact on adolescents' body image and self-image. Females, more than males, viewed social media's impact on their self-image as negative, and most of these negative descriptions were related to unattainable ideals and idealized body representations. A deeper insight into the male's experiences through focus groups revealed that males do experience body-image pressure but that it is rarely talked about. Further, from both datasets adolescents take measures to pursue social media for its positive content, whereas, from the interviews, young males used body-related content on social media as motivation to work towards building their ideal masculine body. Body positivity was talked about in the interviews, where females viewed this as positive, whereas males could not relate to this trend. These findings highlight the need for novel strategies to shift focus away from appearance-related self-images and reduce body-image pressure on social media.









**Data availability statement:** All relevant data are within the manuscript and its Supporting Information files. Due to participant privacy, the qualitative data material from the focus groups will not be shared. The online data is anonymous and will be shared as supporting information alongside this revision.

**Funding:** This work was carried out with financial support from interdisciplinary research funding provided to MKT by Kristiania University of Applied Sciences. The funders had no role in study design, data collection and analysis, decision to publish, or preparation of the manuscript.

**Competing interests:** The authors have declared that no competing interests exist.

## Introduction

Parents, peers, and social media are the primary drivers of sociocultural influence among young people, meaning they learn, develop, and grow in these environments [1]. Social media is considered the most pervasive and persuasive of these drivers. This is due to applications being largely visual platforms, which ceaselessly expose young people to a wide assortment of idealized images that reflect appearance and performance ideals as well as other personal characteristics [2]. This exposure to idealized images makes adolescents particularly vulnerable to social influences since they are in a formative age when their identity is formed, in part, through social comparisons [3,4]. Another worry related to social media is the personally targeted content that may discourage critical reflection due to adolescents' limited awareness of social media functionality [5], which could make shared posts appear more realistic and representative than they are. Moreover, the continuous gratification provided to social media users by likes, comments, and followers may be misinterpreted as reliable guides to what others like and thereby reinforce idealized self-presentations [3,6]. Social comparison could add to the critical developmental stage when the bodily changes of puberty are on the rise, thus impacting on adolescents' body-image and self-image [7, 8, 9].

### Body image and self-image

Body image and self-image are related but distinct concepts. As the name implies, body image refers to an individual's perceptions and attitudes toward their own body, in particular its shape, size, and aesthetics [10]. Self-image can be simplified as a broader view of oneself, including physical appearance as well as personal traits and social roles; it also includes introspective components such as self-concept, self-knowledge, and self-memory [7]. Hence, both concepts encompass several dimensions that may change over time, meaning that there are differences in how much emphasis people place on their distinct characteristics [11].

Adolescents' body-images and self-images seem to develop somewhat different throughout adolescence. The self-images of both males and females do not necessarily arise at a fixed stage in life, however, it seems that adolescents become increasingly focused on their own personal traits and attributes during this life stage [7]. Further, female adolescents seem more prone than males to relate their self-images to physical appearance, whereas males more often relate their self-images to sports or functionality [7]. Where the former characteristic points to a strong integration of body image in females' self-image, the latter may reflect performance ideals among young males. Nevertheless, a qualitative study of 13- to 17-year-olds found that sports and functionality were closely tied to body ideals also in males, particularly muscularity [12]. Thus, it seems that while many adolescents place importance on their body appearance, males and females have different motivations for doing so.

While self-image tends to improve steadily [8], body image tends to exert negative effects on adolescents [9,13]. For females, reaching puberty seems to reflect the onset of this development, whereas puberty does not seem that impactful on males

[9]. Body image and body-image concerns continue to develop throughout adolescence, but typically with differences between males and females. Healthy female adolescents are generally more concerned about their weight and shape than are males, with body dissatisfaction increasing with higher BMI [9]. Among 15-year-old females, body-image concerns have also been related to the drive for thinness [13]. Somewhat different findings have been reported for males, with older underweight adolescent males being at greater risk of developing body dissatisfaction compared to those who are younger [9]. The latter finding might reflect a desire to be muscular that increases toward the end of adolescence [9]; it also aligns with arguments presented by Jarman et al. [14] on how young males are more inclined to internalize muscular ideals when they use social media more for purposes related to appearance and fitness. However, contradictory findings from a qualitative study indicate that the influence of peers on body-image perceptions becomes weaker as male adolescents age [12].

Research on the influence of social media has predominantly focused on adolescents' body image, with comparatively less attention given to its impact on self-image, and this focus has been more pronounced for females than for males [15]. Existing literature is largely dominated by correlational studies, with a relative scarcity of qualitative research [15]. Additionally, methodological limitations persist, such as concentrating on single social media platforms and failing to account for the evolution and diversity of emerging platforms.

A recent study utilizing a seven-day diary approach to assess screen time across multiple social media platforms found no significant associations between social media usage and body image concerns among young females, who comprised the majority of the sample [16]. These findings underscore the need to increase the representation of male participants and to prioritize investigations of content-driven engagement with social media, rather than relying solely on screen time as an indicator of body image dissatisfaction.

### Body ideals and social comparisons

In the original theory of social comparisons by Festinger [17], individuals are said to have a drive to self-evaluate both their opinions and abilities. These types of evaluation necessitate comparisons with others, particularly with individuals who are not too dissimilar, although self-evaluations of abilities tend to be upward comparisons. Wood [11] further proposed that when someone is unfamiliar to oneself, upward comparisons tend to be driven by a motivation to learn from others. However, the impact of comparisons with others seems to increase as individuals become better acquainted or share common attributes. This might cause upward (but quite similar) comparisons to be considered a threat to rather than an inspiration for adolescents' self-improvement [11].

The theory of Festinger [17] has since been extended to self-evaluation of physical appearance, for instance in comparisons with peers and models, and a crucial aspect is that the latter group is only relevant for comparisons due to cultural beauty norms established by social media [18]. These beauty norms have been linked to certain muscular ideals of male and female adolescents [14], where the male body should be big and muscular, including with a six-pack, the female body should be muscular with no six-pack, and slim while at the same time being curvaceous [19]. These body ideals aligns with the appearance focused self-images as described by Hards et al. [7]. Sociocultural theories suggest that the idealized beauty and body images that people strive for based on exposure to social media cannot be attained in a healthy or sustainable way [14,19], and so any comparison made is likely to lead to a negative evaluation [20].

### Body positivity

Recent findings suggest that a novel trend is counteracting some of the negative impacts of social media on body image. This trend is called the body-positive movement (or body positivity), which is thought to have emerged as a reaction to the never-ending stream of idealized images on modern social media platforms [21]. Further, body positivity has been described from a philosophical point of view as the transition from suppressing body shame to embracing body pride [22]. In short, body positivity is predominantly observed in the form of tagged and shared posts on social media platforms (e.g., Instagram, Snapchat,

TikTok) that depict diverse body sizes and appearances [23]. The trend has therefore highlighted the acceptance of ordinary and diverse body types. Early investigations of this topic found that body positivity can change self-evaluations of young people [3] and contribute to a positive body image [21]. However, a few studies have moderated these findings [24,25], with Mahon and Hevey [24] even reporting that adolescent social media users find all body-related content to be harmful. Further, one criticism of body positivity has been its almost exclusive focus on body appearance in social media posts aimed at cultivating a positive body image [21]. Moreover, there has been little research into the potential benefits of body positivity for males [21].

The multiplicity of sociocultural and technological factors involved in the relationships between social media, body image, and self-image is a strong argument for extending the current body of knowledge. Therefore, the aim of this study was to investigate social media influences on male and female adolescents` self-image and body-image.

## Materials and methods

### This study analyzed qualitative data obtained in focus-group interviews

and a cross-sectional survey where the former was generated as part of a larger project on social media habits among adolescents [5]. The triangulation of different methods and data from different participants gave us a broader understanding of the concepts of body image and self-image and how the two concepts relate. Approval to conduct the focus group interviews was initially sought to the Norwegian National Research Ethics Committee. They concluded that it did not require further evaluation and only needed approval from the Norwegian Centre for Research Data due to the collected personal information (reference 644850). In Norway, only medical and health research projects require ethical approval; since this survey was conducted anonymously, ethical approval nor a data handling evaluation was required.

### Focus-group interviews

**Recruitment and procedure.** Adolescents who were students at a secondary school in an urban municipality of southeast Norway (Ringerike) were eligible to participate. Ringerike is considered a medium sized municipality, and has approximately 30,000 inhabitants. After contacting schools in the area to determine their willingness to participate in this study, one lower secondary school (final year) and one higher secondary school (all levels) agreed to participate. One coordinator from each school had overall responsibility for recruitment. Recruitment was performed in randomly chosen classes. If the number of adolescents exceeded the planned size or number of focus groups, a random draw was performed. Those who were not drawn to participate were put on a waiting list. Four adolescents were replaced with those who withdrew before the interviews had started.

A sample of 48 participants (20 males and 28 females) was divided into 8 focus groups based on grade level, with each consisting of 6 adolescents. Focus groups were chosen as the investigation method since we believed that the adolescents would be more willing to share their thoughts and feelings in a group setting than in one-on-one interviews [26]. Further, to create a safe and comfortable environment for the interviews, students in the same classes or at the same class level were selected for inclusion in each group. Further, the group size of six was chosen to facilitate all group members having the opportunity to speak and participating actively in the discussions [27]. Creating both same-gender and mixed-gender groups to investigate gender differences allowed us to explore various social contexts based on the group composition [26].

Twenty-four adolescents in their final year at lower secondary school participated (age 15–16 years), who were assembled into two male and two female groups. Twelve adolescents in grade 1 from upper secondary school participated (age 16–17 years), who were assembled into one male and one female group. Two mixed-gender groups were assembled from adolescents in grade 2 (age 17–18 years) and grade 3 (age 18–19 years), which included one male and five females in each group. One male participant in the latter group was 24 years old, which was due to general education rules of the Norwegian school system that allow enrollment until the age of 24 years. Across the sample of 48 participants, the self-reported average daily use of social media was 4.8 hours, with no observed difference between genders.

Prior to the interviews, the interview questionnaire was tested in a group of adolescents to ensure that the prepared questions were both relevant and easy to understand [5.] The interviews were performed in a private room at each school from 19th April to 1st June 2021. The location was chosen for convenience, but it was also a place of familiarity, which increased the feelings of safety and neutrality. Before the interviews, signed consent forms were obtained from the adolescents and from the parents of those younger than 16 years. The participants were also verbally informed of their voluntary and ethical rights at the start of each interview, in addition to the study aim. Two female researchers were present at the interviews: one (A.R.B.) was experienced in qualitative interview techniques and led all the interviews, at the time employed as a research assistant, while the other researcher (M.K.T. or R.E.) took notes on speaking and nonverbal behaviors; neither researcher was familiar with the participants. The interviewer made a conscious decision to be an active listener and to follow any cues provided by the participants during the interviews. Both researchers and participants were positioned so that everyone could see and interact with each other. To avoid any bias, the interviewer strived for an open discussion on the topics of social media and body image, while paying attention to all participants. Silent participants were approached by the interviewer directly so that all of them had an opportunity to speak. Interviews lasted 60–90 minutes and were recorded using a secure Dictaphone application created with nettskjema.no, developed and hosted by the University of Oslo (nettskjema@usit.uio.no). Transcripts, recordings, and signed consent forms were stored according to general data protection regulations. All participants were given a gift card valued at about 30 USD as compensation for their time and they were given the opportunety to comment and correct their trancripts. However, none of the participants provided their feedback.

The interview addressed different topics related to social media while following a semi-structured interview guide [5]. After being introduced to the general themes and ground rules for discussion, the participants answered the following questions about body image: "In what way would you say that social media influences you?" "Based on what you look at, what makes you feel better, and why do you think that is?" and "Do you believe that social media contribute to body-appearance pressure?" For the latter question, body-image pressure was brought up when asking the first two questions. Thus, this latter question was posed more as an affirmative question. The interviewer also posed the following question: "Based on what you look at on social media, what makes you feel worse, and why do you think that is?" However, no new responses were received since this was a follow-up question.

**Analysis.** The qualitative approach was informed by a phenomenological worldview [28], where we explored adolescents` experiences with social media and body image. The transcripts were written verbatim and supplemented by the notes taken on speaking and nonverbal behaviors, with fictitious names used to ensure anonymity. We carried out a thematic analysis to identify repeated patterns of the influences of social media on body image [29]. An inductive approach was used due to the social contexts of the experiences of adolescents. The social context of the study was also kept in mind while striving to capture similarities and differences within and between the various age and gender groups. All data were coded and themes were categorized using NVivo [30].

We followed the six steps proposed by Braun and Clarke [29] for our analysis, which was consistent with our previous approach [5], as illustrated in Figs 1 and 2. Steps 1 and 2 included familiarization with the data and coding (performed by A.R.B.), step 3 included identifying themes by merging codes (performed by A.R.B. and M.K.T.), step 4 involved all authors reviewing and approving the initial themes, step 5 involved defining, naming, and rereading the final themes by all authors to ensure consensus, and step 6 included selecting suitable statements to illustrate the final themes (performed by all authors). All researchers agreed on the statements that would best describe each theme while considering the study objectives. The statements reported here were translated from Norwegian to English and edited slightly to improve readability, while remaining as close to the original statements as possible. The Consolidated Criteria for Reporting Qualitative Research [31] were followed.

### The survey

**Recruitment and procedure.** The focus group interviews were succeeded 6 months later by a national survey administered online during 14th December 2021 and 12th January 2022 by the Norwegian branch of the market agency

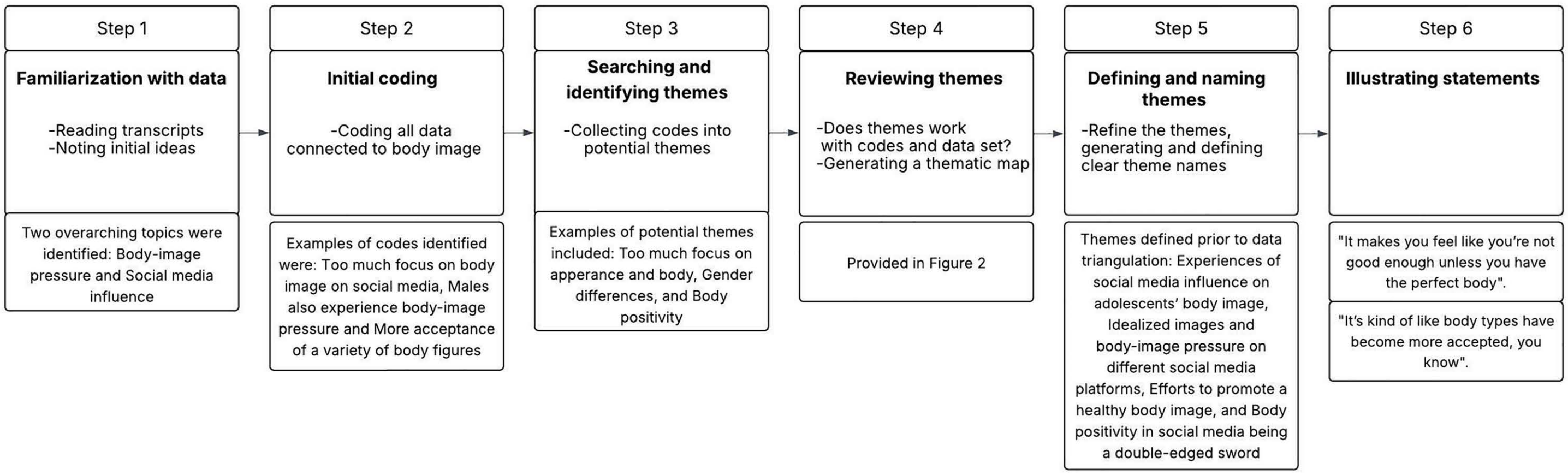


**Fig 1. Phases of thematic analysis.** Following step 5, the themes were further refined through data triangulation, supplemented by a questionnaire incorporating measures of self-image.


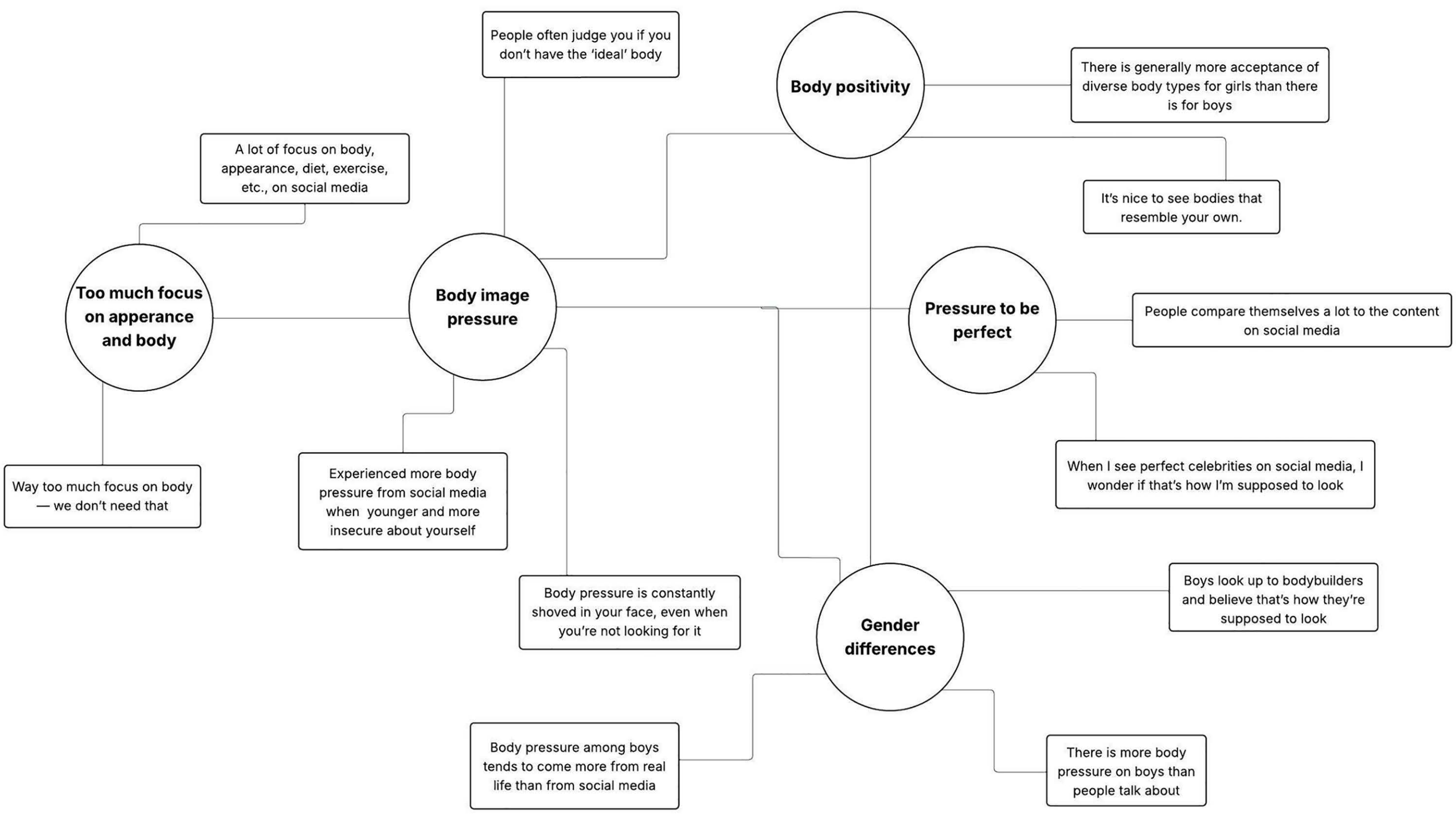


**Fig 2. Initial thematic map illustrating the five main themes (circles) and their corresponding codes (squares).**


Kantar, who recruited respondents through popular social media platforms. This second study's purpose was to gain information about self-image, the broader aspect of body image that the focus groups did not discuss. Apart from age (16–20 years), gender (identifying as either male or female), and consenting to participate (electronicly through the survey), no other inclusion criteria were applied. The lower age range was set based on the practical requirement to receive consent, and the upper age range was chosen due to the Norwegian school system offering vocational training and other specialized forms of education that take 4 years (rather than 3 years) to complete. Although 20-year-olds would not typically be defined as adolescents, the Norwegian school system includes them as secondary-school students and this places them in the study's target group. Adolescents who saw the social media posts and wanted to participate would click on a hyperlink that opened the survey in their browser. The required sample size was not calculated a priori.

The survey comprised demographic items, including gender, 15 questions addressing social media usage patterns (e.g., time spent, platforms used, and types of activities), perceived influence of social media relative to time spent, participants' knowledge and awareness of personalization, and one open-ended item concerning self-image: "How would you describe the influences of social media on your self-image?" We deliberately chose not to mention body image in order for the participants to reflect on the wider meaning of body image as part of their self-image.

There were 339 respondents who completed the full survey, most of whom had completed the education levels expected at their respective ages. The agency's online recruitment process strived to ensure that respondents came from all national regions, along with an overrepresentation of the southeastern regions in line with the overall population distribution. Prior to performing the analysis, 26 respondents were excluded since they fell outside the defined age range or did not report their gender. Among the remaining 313 respondents, 127 answered the open-ended question related to self-image; 3 of these were excluded due to unrelated or incomprehensible responses. This resulted in 124 open responses being eligible for analysis, corresponding to a response rate of 39.62%. The respondents were aged 18.03 ± 1.39 years (mean±SD), and their age and gender distributions are listed in Table 1. Participants self-reported an average daily social media use of 4.48 ± 2.30 hours, with females reporting 4.74 ± 2.24 hours and males reporting 4.16 ± 2.34 hours.

**Analysis.** The responses to the open-ended question were pooled into three categories: positive, negative, and neutral/mixed. The first two categories reflected positive or negative feelings about the influences of social media on self-image, whereas the neutral/mixed category reflected either a combination of both positive and negative feelings or that the participant had no opinion about the topic. This categorization was first performed by one researcher (A.R.B). The two other researchers (M.K.T and R.E) then independently performed the same coding, before all three agreed upon which category each response would fall into. The categorization was done qualitatively and based on the subjective evaluations by the authors. After a joint evaluation and revision, complete unanimity was reached. Two statements from each category were then identified from the data material to best represent the various feelings of both genders. The statements were then translated from Norwegian to English and edited slightly to improve readability. The data were also quantified by summing the number of statements in each of the three categories and converting them to percentages. Differences between males and females in all three categories were analyzed using chi-square tests and reported as numbers with percentages and probability values, with $p < .05$ considered significant. To increase the statistical power of the analysis, all age groups were merged.

**Table 1. Age and gender distributions of the 124 survey respondents who provided eligible responses.**

| | 16 years | 17 years | 18 years | 19 years | 20 years | Total |
|---|---|---|---|---|---|---|
| **Females** | 17 | 13 | 14 | 13 | 16 | 73 |
| **Males** | 6 | 10 | 16 | 10 | 9 | 51 |


### Triangulation of data

Triangulation of data from two different methods and two different populations was done to develop a comprehensive understanding of adolescents' self-image and body image [32]. Identifying the themes from the focus groups and the categories from the survey, all authors synthesized the results, looking for differences and similarities in our findings [33].

## Results

Regarding the first theme, *'Experiences of social media influence on adolescents' self-image and body image,'* the results indicated that most adolescents perceived the impact of social media as predominantly negative. Their descriptions of self-image were frequently associated with unattainable ideals and idealized body representations, patterns recognized from the focus group analysis.

Regarding the second theme, *'Efforts to promote a healthy self-image and body image,'* survey responses demonstrating positive strategies aligned with focus group findings, wherein some adolescents reported actively seeking positive social media content to enhance their self-image and body image. Notably, focus group discussions provided a more nuanced understanding, particularly regarding male participants' engagement with positive content.

Finally, experiences related to body positivity emerged exclusively in the focus groups and were not corroborated by the survey data. Additional illustrative statements are provided in Supplementary S1 File.

### Theme 1: Experiences of social media influence on adolescents self-image and body-image

The qualitative categorization of the survey statements fell into three categories: 55 respondents (44.4%) reported that social media had a negative influence on their self-image, compared with 20 (16.1%) who provided positive reports and 49 (39.5%) who reported mixed or neutral feelings (Table 2).

More females than males (56.2% vs. 27.5%) expressed that social media had negatively influenced their self-image ($X^2 = 6.29$, $p = .01$). No differences was seen for the negative or mixed/neutral feelings between gender. Most of their described self-images were related to appearance and unattainable body ideals, giving the impression that body image plays a significant role in adolescents' self-images, more so for females than males. Mona, a 20-year-old female, explained this negative influence as follows:

> [Social media] has a bad influence. There will always be someone who looks better, does
>
> it better, has it better, and is living 'my dreams.' I'm more hesitant with my decisions because of social media, and [as a result] my self-image is weakened.

Although not explicitly referring to body image, Pia, aged 16, said, "My self-image is influenced by ideals I know I cannot fulfill." From the interviews, there appeared to be a pattern of gender differences in how they experienced body ideals. This pattern of agreement was shared within and between interview groups regardless of gender and group composition. When students in grade 2 from higher secondary school were asked in the interview if they believed that social media contribute to body-image pressure, Suzanne (aged 17–18 years) said: "You sort of get it right in the face. Like, if we didn't

**Table 2. Positive, negative, and neutral/mixed feelings regarding the influences of social media on adolescents' self-image.**

| | Females ($n = 73$) | Males ($n = 51$) | Total ($n = 124$) |
|---|---|---|---|
| **Positive** | 8 (11.0%) | 12 (23.5%) | 20 (16.1%) |
| **Negative** | 41 (56.2%) | 14 (27.5%) | 55 (44.4%) |
| **Mixed or neutral** | 24 (32.9%) | 25 (49.0%) | 49 (39.5%) |


have social media, it would not have been so easy to feel it [body-image pressure].” The experiences of body-image pressure were often attributed to how social media present the perfect life and body. Several of the participants expressed that the current body ideals are unattainable for many. For example, Elizabeth (aged 15–16 years) said: “It’s not enough to just be thin. You are supposed to be thin, but also curvaceous.”

Looking more closely into the social contexts of the different groups, males shared their experiences with body-image pressure more openly and in more detail when they were in same-gender groups. They expressed that body-image pressure occurred among male adolescents, but emphasized how they believed that their experiences were different from those of females. According to Sam (a male aged 16–17 years): “I feel like there’s more body-image pressure among boys, it’s just not talked about that often,” to which several fellow interviewees either nodded or said “yes” in agreement. The participants offered an additional explanation for this gender difference, where young males experience more body-image pressure from peers and the physical environment, while females are more prone to experiencing this type of pressure on social media. The males also expressed that there were different “rules” between how males and females portrayed ideal body types, and that these caused an imbalance between genders. Sam said the following:

> I feel like, if a girl talks about the man of her dreams, it’s all good that they [females] think that he should have a six-pack and be super handsome. But if a boy does it [describe their dream woman] and says that she should have a nice bum and large breasts, then it’s wrong.

### Theme 1.1 Idealized images and body-image pressure on different social media platforms

Some participants mentioned in the survey that Instagram influenced their self-image. During the interviews, Instagram was again mentioned, but explicitly in relation to how social media contributed to body-image pressure. Although most prevalent on Instagram, Snapchat and TikTok was also mentioned as sources of body-image pressure. The adolescents talked about the desire to receive feedback via comments and likes, while some said that there were alternative social media platforms that counteracted this desire. The analysis of body-image pressure related to social media platforms revealed no clear response pattern across the different group compositions. Further, the adolescents elaborated on this topic more objectively compared with the subjective feelings they described about the impact of these platforms. Peder (aged 16–17 years) answered as follows when he was asked about whether social media contributed to body-image pressure:

> I believe that there is a lot of body-image pressure on Instagram. Girls look up to a model and believe that is the way they are supposed to be. And we boys look up to body builders with nice and fit bodies, and everything is supposed to be perfect.

The desire to receive likes and comments is also fueled by social media usage and was mentioned by the male participants in our study when asked about what makes them feel better on social media. Christoffer (aged 15–16 years) stated: “It is a good feeling if you for example post something on Instagram, and you see that you get many likes. But it can also [backfire] (…) if you get no likes all of a sudden.” This feeling was shared as a general comment and was not specifically related to body image. Moreover, some of the females appreciated the VSCO social media platform because they experienced less pressure when using it. Suzanne explained her experience with VSCO: “You don’t have to worry about the pressure with followers, likes and comments and such, because it is not possible to see how many likes a person gets on a post [on VSCO as compared to Instagram].”

### Theme 2: Efforts to promote a healthy self-image and body image

According to the survey, some adolescents actively pursue social media for its positive impact. Hayley, a 20-year-old female, addressed her self-image in this context:

> My self-image has improved from social media, but that is because I have actively pursued positive content (…). Of course, some places [on social media] are not positive, but this wide assortment teaches you to ignore [content] and be critical.

This way of pursuing content on social media was also discovered from the interviews. Among mainly the younger male participants, body-related content on social media was used as a motivation to work towards building their ideal masculine body. They mentioned this practice spontaneously before body image was specifically addressed in the interviews. It was particularly interesting that we observed that the males in the group from the same class seemed more alike in their responses and more hesitant to disclose opposing thoughts and experiences. Tom said: "If we [males] see someone (…) that looks really good, then I would use that as a motivation [to do something about it]," which some of his peers agreed with. During the same interview, Simon expressed similar sentiments: "Girls can often get more depressed because they don't feel they are perfect. But boys will kind of make an effort to become the best version of themselves." A heated debate started when the other group of younger males coming from three different classes discussed using body-related content as an inspiration to get fit, with one male participant expressing that motivation deriving from unrealistic body ideals was unhealthy and unsound. Kevin (aged 15–16 years) expressed concerns about this practice: "Well, that kind of motivation is not that good because then your goal is to be like them [influencers], and that is not your goal. You are not going to be like them." The findings from these two male groups may reflect prior acquaintances, with males from the same class group possibly withholding opinions that were not consistent with the shared opinions in that group.

The practice of avoiding content was also found from the survey respondents, but mainly among the females describing how they had deleted social media accounts or stopped following people. Petra, a 20 year old female expressed:

> I used to have a poorer self-image, and it depends on what I see on social media. So, in recent years, I have stopped following those who make me feel I can never live up to perfect standards and instead follow inspiring people.

During the interviews, a few of the older participants mentioned the practice of unfollowing or not following specific accounts or people on social media. Markus talked about how he had deliberately chosen not to follow accounts that posted content that might portray bodies while also pointing out that many adolescents do not make active choices for themselves to reduce their exposure to this type of content: "(…) but I think that many people do it [follow people that post body-related content]. A lot of people get influenced by celebrities that they follow, in relation to body-image pressure." Suzanne shared the practice of not following such accounts and elaborated during the same interview: "I don't feel I get very influenced, on body matters and such. I don't really follow many people that post these things, and I don't really seek it out."

### Theme 2.2: Body positivity in social media being a double-edged sword

Topics related to body positivity were discussed by the participants in the interviews with great interest. This came up spontaneously in response to more general questions on the influences of social media, as well as from the specific question of how social media contribute to body-appearance pressure. Body positivity was seemingly experienced differently by males than females, with the females reporting greater appreciation of body positivity, while the males were more skeptical. However, it must be noted that it was not possible to capture the male voices related to body positivity from the two mixed-gender groups.

Several of the oldest females recognized body positivity as an emerging trend that they appreciated. Like Maria (aged 18–19 years) said:

> There's been more body positivity now in the last few years, especially on platforms like TikTok showing different kinds of people. Like bigger girls, smaller girls, boys that are more feminine. So yes, it [body positivity] is a big thing.

The notion that this was a positive trend was also shared by the younger participants in our study. Jenny (aged 16–17 years) said: "Personally, I get really happy about body positivity stuff. That is, it is ok to be yourself and that you are good enough as you are, in a way."

Our finding of body positivity being considered a positive trend predominantly came from the females. As Adrian (aged 16–17 years): said: "Recently I have seen more female models that do not fit the Barbie look (…). But that only goes for the girls. It's not like you ever see a fat man on Instagram, right?" During the same interview Peder (aged 16–17 years) was in agreement: "For example, in Calvin Klein commercials you would never see a guy with a somewhat larger belly, everybody has a six-pack." The males also expressed their negative perceptions about body positivity, such as how it promotes obesity and unhealthy lifestyles. Further, although the adolescents noted that body positivity seemed to introduce some body diversity, they still found that there was too much focus on appearance. Jenny said: "It's like, bodies have become more accepted. But there is just as much body-image pressure (…)." The same sentiments were expressed by the older females.

## Discussion

### How social media influence adolescents' body image and self-image

This study of self-image and body image has revealed a social media landscape abundant with content that portrays unattainable ideals and idealized body representations that are burdensome to many adolescents in Norway. The survey responses indicated that social media`s influence on adolescents' self-image was more negative for the females as compared to the males. The female self-images were also more often related to body ideals or to their own appearance and body-image. This aligns with the study of Hards et al. [7] where females were more likely than males to describe their self-image as being related to appearance. The observation that adolescents' self-image seems to be closely tied to unattainable ideals, their own appearance and their appearance ideals, thereby increasing their vulnerability to upward comparisons, is a concern [17]. Previous population-based studies support our findings of females being more negatively influenced by social media than males [34]. Our survey findings also indicate that females use social media more than males. In Norway, Snapchat, TikTok and Instagram are generally used more by female than male adolescents, in that order [35], implying that young females are more exposed to content that could give rise to negative feelings. Algorithms on social media platforms determine what the user will be exposed to, and this selection process may also vary with gender [19]. In the school interviews, Instagram was mentioned repeatedly as a platform that portrays unrealistic beauty ideals, making it a source of body-image pressure through upward comparison [17]. However, considering that Instagram is used by 81% of adolescents aged 16–19 years, its observed influence may primarily reflect its widespread adoption [35], rather than any platform-specific characteristics. This observation was consistent with Hjetland et al.`s [36] finding that Instagram showed attractive photographs that do not necessarily reflect reality, at least not among Norwegian adolescents aged 15–18 years. However, there seem to be gender-related differences in how social media influence body image. Our interview data suggested that males talk less about body-image pressure on social media. This could be due to stigmatization, but may also simply be due to males being less concerned, as was mentioned by some of the male participants in the interviews. However, another argument is that males avoid expressing their emotions in order to adhere to acceptable masculine behaviors [37]. The social contexts in the interviews might also have been important, with the males being reluctant to disclose their true feelings in mixed-gender groups or mixed classes when those feelings did not match the shared opinions and feelings of others [26]. The absence of disclosing emotional vulnerability might have resulted in underreporting of the numbers of males who experienced body-image pressure. It also seems that different rules apply to males and females, for example in how females can talk about the "perfect" male body and focus on characteristics that are difficult to change, such as height and body composition. Further, the younger males in our study experienced more body-image pressure from their physical and social environments than from social media, possible because their relative closeness to peers and family members makes them more relevant than social media for comparisons [11].

These younger males might also be vulnerable to social comparisons during transitional phases at secondary school [12]. Simultaneously, receiving support from peers and family members has been shown to be important for the development and maintenance of a positive body image in both genders [3,12]. Nonetheless, the understanding of male body-image pressure is far from complete, and we stand with others in calling for more qualitative studies [24].

Goodyear et al. [38] found that health-related information, such as on diet and exercise, can positively affect the health-related behaviors of young males and females. However, 24% of the younger adolescents in the study by Goodyear et al reported that the health-related content they see on social media had a negative impact on their health-related behaviors. Although not stated explicitly, it was clear from our interviews that adolescents` views of their own bodies were influenced by social media (often negatively) for both females and males. The participants also expressed how they compared themselves with people they saw on social media, supporting the theory of Festinger on social comparisons [6,17,18]. This is consistent with our survey revealing the negative feelings experienced when the participants viewed unattainable ideals on social media. According to Wood [11] these negative feelings appear when people compare themselves to others to whom they share some common attributes. Thus, the unattainable ideals viewed on social media may work as a threat to adolecents self-image, considering the proximity that follows repeated exposure. Although we did not look into adolescent self-image over time, it seems that among 16–20 year-olds, their self-image is closely tied to apperance, especially among females [7]. As a consequence, upward comparisons may contribute to form the self-image of adolescence [8]. The females further elaborated in the interviews that the pressure from social media was due to the conflicting ideals of being thin but also curvaceous, while the males perceived pressure to be fit and muscular, similar to the reports of Jarman et al. [14] and Steinnes et al. [19].

The finding of Jarman et al. [14] that muscular-ideal internalization is positively associated with body satisfaction and well-being could support our findings of the younger males using exercise content they viewed on social media as a motivation to work out. Wood [11] reported that highly motivated persons may find social media inspiring due to upward comparisons being perceived as attainable ideals, and thereby avoiding feelings of inferiority. However, we did not obtain data on the actual exercise patterns of our interview participants and hence we could not determine if their training was excessive. We also did not investigate other health-related behaviors such as dieting or taking supplements. Further, exercise videos on Instagram may encourage male users who already have a negative body image to transform their bodies to be more similar to what they see on social media [39]. Supporting sociocultural theories, there is increasing evidence of thin and muscular internalization among adolescents and young adults being positively associated with unhealthy ideals [40]. Since body-image concerns tend to develop and increase throughout adolescence, these findings could be of concern [9].

### Different social media platforms, adolescents`social media practices and their influences on body-image pressure

The reports of participants regarding the influences of social media on body image suggest that platforms such as TikTok and Instagram are particularly adept at targeting young people with idealized images related to appearance or other characteristics. Further, the methods that TikTok and Instagram use to target their users based on demographics and online behaviors may uphold and reinforce stereotypes [19], such as those of gender and beauty. Social media targeting is not necessarily all bad, since personalized content might be more interesting and relevant to the user [5]. However, we previously found that adolescents appear to be relatively unaware of how personalization in social media works [5]. Thus, the continuous exposure to unrealistic ideals and unattainable beauty and body ideals may have resulted in both the male and female participants in the present study becoming even more prone to social comparisons. As a consequence, these unrealistic ideals may impact adolescents' self-image and body-image.

The practices of deleting or not following people or accounts that contribute to a poorer self-image and body-image were mentioned by some of the adolescents. However, from the interviews most of them did not seem to take actions aimed at reducing body-related content on social media, even when they were conscious of it. Our own recent findings

suggest that this acceptance could be due to the perceived benefits of social media [5]. Hjetland et al. [36] stated how the users of Instagram reported a desire to receive likes and comments, similar to the findings of the current study. This acknowledgement of idealized self-presentation may be another explanation for why adolescents tend to continue using platforms that promote body-related content [4,6], in addition to the never-ending stream of entertainment [4]. VSCO was also appreciated, but for a different reason; it does not display numbers of likes or followers, which was perceived as reducing body-image pressure. Although these are positive findings for VSCO, this platform was only mentioned twice, both times by female participants, suggesting that the impacts of less-approval-oriented platforms can vary with gender. Although social media were reported to be the main source of influence, the mentioned social comparisons with family and peers must also not be overlooked [6].

### Body positivity and how it influences body-image pressure

From our interviews, several of the females in the oldest age groups regarded body positivity as an emerging and appreciated trend. Although in its infancy, research has indicated that content and conversations related to body positivity may lead to an improved self-image and less body-image pressure [3,21,25]. These findings coincide with our participants expressing that body positivity made them see themselves in a better light. Further, the females in our study compared themselves with models and influencers on social media more often than did the males; the body-positivity trend with its many influencers may therefore have had a stronger impact on females. However, it was also observed that some of the males did not feel that body positivity was directed at them; rather, they felt that social media posts did not share the same diversity of body-related content for males as they did for females. It appears that research in this area has mostly addressed females [3,21,25], which might be due to social media usage reportedly having a stronger negative impact on females than on males [34].

There have been reports that body positivity promotes obesity and unhealthy lifestyles, which was a concern among some of the younger males in our study. However, there is no evidence in the scientific literature supporting these reports, and they probably instead reflect weight biases associated with unhealthy lifestyles [41]. Cohen et al. [23] and Mahon and Hevey [24] also found that young people tend to view all body-related content as damaging. Both the younger and older females in our interviews raised this concern and expressed that the body-positivity trend was just as focused on appearance as social media are in general. Cohen et al. [21] proposed using quotes and illustrations to promote conversions about body positivity in order to move the focus away from appearance and toward positive body images. Another suggestion has been to aim for body neutrality by changing the value of beauty norms from physical appearance to a broader conceptualization of beauty, including body functionality and inner positivity.

### Implications for practice

The findings of this study have several implications for practice. Although social media seem to impact both gender in terms of self-image and body image, this study provides valuable data on the influences of social media on males, especially younger males who may be more reliant on unhealthy and unattainable role models on social media when performing comparisons. It is also important to be aware that although some trends have positive effects on many people, such as body positivity, their influence can vary with gender and age. One useful approach could be universal and interactive social media literacy programs that strengthen the critical stances of adolescents regarding the influences of social media [5,15]. Importantly, these programs must be designed while allowing for differences with gender and between populations with varying levels of social media usage. Further, a previous study from our research group highlighted the need for increased attention from parents and teachers on improving social media awareness among adolescents [42]. Additional strategies are required to mitigate the pressures induced by social media portrayals of idealized images, particularly those emphasizing body-appearance ideals. One possible approach would be to focus on body functionality rather than the appearance of the body, while another would be to promote social media platforms that hide number of likes and

comments. Considering the importance of body image, reduced focuses on body appearance and social approval would probably be beneficial for the overall self-image of users.

## Limitations

The design of this study made it possible to assess social media influences on adolescents' self- and body image. Although triangulation of data adds to the comprehensive understanding of these two concepts, the age and gender composition in the the two samples may be problematic. First, the age range was slightly higher for the survey respondents than for the interviewees. Second, while the gender compositions were similar across the two populations, more females participated across the entire age span. Although we strived for the engagement of all participants throughout the focus groups, it is possible that we missed important information, especially from the few male participants in the mixed-gender groups. Thus, the social context of being a single male in a group of females [26] meant that we were not able to capture the male interviewees' voices related to body appearance and body positivity. Thus, these interpretations must be made with caution, and future research should consider the recruitment of older male adolescents. Importantly, the inclusion of a male interviewer could also contribute to male adolescents more openly sharing their experiences with social media influences.

Another limitation pertains to the qualitative categorization of the survey respondents. Although complete unanimity was achieved, potential bias in subjective categorization cannot be ruled out. Nevertheless, the collaborative review conducted by researchers from diverse academic fields—public health, psychology, and social media studies—enhances the credibility of this subjective interpretation.

Further limitations could have arisen from the questions asked. The key questions from the survey and the interviews centered on the influences of social media on self-image and body image. Thus, the emotions expressed may have been induced by these specific questions. Nevertheless, we strived to keep the questions open to get a nuanced and detailed impression of positive and negative emotions with self-image and body image on social media. From the interviews, both males and females brought up the issues of body image pressure and body positivity before the researchers presented them. In the survey, a greater proportion of females than males described their body image as an integral component of their self-image. Many respondents also identified unattainable ideals as a source of influence. Although adolescents tend to associate their self-image with appearance to a greater extent than adults [7], definitive conclusions cannot be drawn as to whether these ideals constitute body-related ideals specifically or reflect other personal characteristics or attributes.

For the survey, adolescents were sampled from all national regions. However, the response rate was relatively low, which may introduce self-selection bias, and thus question the generalizability of these results. Further, for the interviews, adolescents were included from one municipality only, meaning that these adolescents may not be representative of all adolescents in the country. Although the time spent on social media was comparable between the two groups and aligned with national data [35], the reliance on self-reported usage introduces potential recall bias. Consequently, the generalizability of our sample to adolescents residing in other regions of Norway, particularly regarding exposure to diverse social media content, may be limited [43]. Importantly, since the study was conducted in Norway, which is characterized by high rates of online activity, our data might not be representative of populations where social media are used less.

We acknowledge that social media may influence adolescents' body image and self-image differently according to ethnicity and gender, including among those identifying as nonbinary, but this was not a focus in the present study. A future research focus could therefore be to investigate how adolescents from various cultures and ethnicities are influenced by specific ideals that may differ from the ideals included in the current study. Further, the cross-sectional design of this study does not allow us to follow adolescents throughout transitional phases at secondary school. Thus, studies exploring the longitudinal impact of social media influence are warranted.


## Conclusion

This study has provided insight into the experiences of females and males with social media and how social media influences their self-image and body image. From our analysis, adolescents' self-images were described in relation to unattainable ideals and idealized body representations, where more females than males viewed the influence of social media as negative. However, there seems to be a pattern of gender differences, as evident from our interviews, where males also experience body-image pressure, but they do not talk about it as often. Further, adolescents do take measures to pursue positive content on social media that could positively influence their self-image and body image. However, for the younger males in the interviews, body-related content on social media was viewed as motivation to work out to become closer to the ideal body. Lastly, the body positivity trend was expressed through the interviews only, and it was appreciated by many of the females in our study. At the same time, it did not impact the males similarly. Given that social media have become an integral part of the lives of adolescents, interventions are required to turn their focus away from idealized images, particularly those emphasizing body-appearance ideals.

## Supporting information

**S1 File. Additional statements.**
(DOCX)

**S2 File.**
(XLSX)


## Acknowledgments

The authors thank the participating secondary schools for their cooperation and help with recruitment, and all of the adolescents who volunteered as participants in the study. The authors also thank Miroslava Tokovska for help in identifying relevant themes and Kethe Marie Engen Svantorp-Tveiten for feedback on previous versions of the manuscript.


## Author contributions

**Conceptualization:** Merete Kolberg Tennfjord, Ashley Rebecca Bell, Ragnhild Eg.

**Data curation:** Ragnhild Eg.

**Formal analysis:** Merete Kolberg Tennfjord, Ashley Rebecca Bell, Ragnhild Eg.

**Funding acquisition:** Merete Kolberg Tennfjord, Ragnhild Eg.

**Investigation:** Merete Kolberg Tennfjord, Ashley Rebecca Bell.

**Methodology:** Merete Kolberg Tennfjord, Ashley Rebecca Bell, Ragnhild Eg.

**Project administration:** Merete Kolberg Tennfjord, Ragnhild Eg.

**Validation:** Merete Kolberg Tennfjord, Ashley Rebecca Bell, Ragnhild Eg.

**Writing – original draft:** Merete Kolberg Tennfjord, Ashley Rebecca Bell, Ragnhild Eg.

**Writing – review & editing:** Merete Kolberg Tennfjord, Ashley Rebecca Bell, Ragnhild Eg.

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
