## [Decision Letter · Decision Letter 0]

6 Aug 2024

Dear Dr. Tennfjord,

Thank you for submitting your manuscript to PLOS ONE. After careful consideration, we feel that it has merit but does not fully meet PLOS ONE’s publication criteria as it currently stands. Therefore, we invite you to submit a revised version of the manuscript that addresses the points raised during the review process.

We look forward to receiving your revised manuscript.

Kind regards,

Annesha Sil, Ph.D.

Associate Editor

PLOS ONE

“This work was carried out with financial support from interdisciplinary research funding provided by Kristiania University College.”

3. In this instance it seems there may be acceptable restrictions in place that prevent the public sharing of your minimal data. However, in line with our goal of ensuring long-term data availability to all interested researchers, PLOS’ Data Policy states that authors cannot be the sole named individuals responsible for ensuring data access (http://journals.plos.org/plosone/s/data-availability#loc-acceptable-data-sharing-methods).

Reviewers' comments:

Reviewer's Responses to Questions

**Comments to the Author**

1. Is the manuscript technically sound, and do the data support the conclusions?

Reviewer #1: Yes

Reviewer #2: Partly

2. Has the statistical analysis been performed appropriately and rigorously?

Reviewer #1: Yes

Reviewer #2: N/A

3. Have the authors made all data underlying the findings in their manuscript fully available?

Reviewer #1: No

Reviewer #2: No

4. Is the manuscript presented in an intelligible fashion and written in standard English?

Reviewer #1: Yes

Reviewer #2: Yes

Reviewer #1: 1. Is the manuscript technically sound, and do the data support the conclusions?

The manuscript is overall good. It presents a compelling exploration of an important topic in adolescent psychology, focusing on both females and males’ ideas and perceptions related to social media. The authors attempt to integrate various sides of the story when it comes to body image and self-image, and try to touch on both. The results are interesting but maybe this section could benefit from further refinement in terms of organization and clarity. The research questions and data do support the conclusions in this manuscript.

*2. Has the statistical analysis been performed appropriately and rigorously?

The statistical analysis of this manuscript appears to be appropriately and rigorously performed. The authors showed attention to detail and provided a strong foundation for the analysis.

*3. Have the authors made all data underlying the findings in their manuscript fully available?

No, the authors declared that there are some restrictions for data availability as well as that they have no consent to share data sampled from the interviews undertaken.

*4. Is the manuscript presented in an intelligible fashion and written in standard English?

Overall, the manuscript presented in an intelligible fashion and written in standard English. There are a few minor grammar mistakes that are not that much important, but should be addressed with a brief review.

*5. Review Comments to the Author

Please use the space provided to explain your answers to the questions above. You may also include additional comments for the author, including concerns about dual publication, research ethics, or publication ethics. (Please upload your review as an attachment if it exceeds 20,000 characters) (Limit 200 to 20000 Characters)

Overall, this paper addresses an important topic as adolescents heavily engage with social media, making it crucial to understand its impact on body image and self-image. By delving into gender differences in this influence, the paper offers a unique perspective. However, improvements can be made, starting with revising the abstract to include background context. While the introduction provides a comprehensive overview, certain points, such as those in lines 49-50 and 55-57, require clarification. The methodology section is well-structured, but has certain limitations, such as sampling bias, the cross-sectional design, and social desirability bias. Also, it is important to note that the paper could have been enhanced by addressing potential confounding variables. Although the identified themes are intriguing, they could be presented more briefly for clarity. The discussion effectively integrates theories like social comparison theory and compares findings with existing literature, encompassing both genders. Additionally, the inclusion of implications for clinical practice enhances the paper's relevance by providing guidance for targeted interventions and support services regarding adolescents' experiences with social media and body image. It is also worth noting that the paper’s focus on gender differences adds depth to the understanding of how social media influences body image and self-image. Such an approach acknowledges that adolescents’ experiences may vary based on factors such as gender norms and societal expectations.

Notes:

- Abstract: missing the background of the topic

- Introduction: overall comprehensive and provides a well-established overview of the topic

- Line 49-50: what is that sociocultural influence? Maybe give examples for the sentence is a bit ambiguous

- Limitation for the methodology section: sampling bias as the adolescents were recruited from one municipality.

- I do not know about using the adolescents first names in the results? Is that okay?

- The themes identified are very interesting, however they are presented in an excessively lengthy manner and could be rephrased to be shorter and more concise.

- Discussion: well-structured and comprehensive discussion – compares and backs up the results of the study with previous literature, included both males and females. Also, there were different theory integration when explaining how social media effects self-image and body image, such as the social comparison theory.

- The implications for clinical practice section in the discussion was important to add as it enhances the comprehensiveness and relevance of the study’s findings, giving an idea for targeted interventions and support services to better understand adolescents’ experiences with regards to social media and body image.

Reviewer #2: Thank you for the opportunity to review this manuscript.

The subject of self-image and its relations to social media is certainly relevant and there are many nuances to be explored within it. However, I think the manuscript has methodological issues that need to be addressed before it can be published.

Firstly, I lacks focus, trying to cover too many objectives and using different types of data that are not triangulated at the analytical and interpretative stages. The paper has currently four objetives: to investigate social media influences on adolescents` body-image; how adolescents view their body image in relation to their self image with emphasis on gender differences; to investigate adolescents’ social media practices; and their experiences with body positivity. My impression is that the thematic analysis was not fully completed, not reaching the stage when sub-themes are systematized into themes.

Secondly, the data collection procedures have limitations that need to be considered in your analysis. The main one refers to the questions that were asked during FG, which induce participants to confirm that SM influences them and their body-image (“How does what you see on social media influence you?”) and some are not open questions (“Do you believe that social media contribute to body appearance pressure?”). The questions regarding SM use do not seem to have been fully analyzed or at least the results are not reported. Two other limitations are: 1. the fact that the FG were done in schools, an institutional scenario where adolescents will not necessarily feel comfortable to talk about themselves and their feelings, which may have contributed to their discourse being more influenced by social desirability; 2. only female researchers conducted the FG, which may have had an important impact on how boys expressed themselves (or refused to).

When we get to the analysis, the results are presented in the form of two separated research questions, reflecting the lack ofdata triangulation. In addition, the first research question contains three questions in ti: "How do social media influence adolescents’ body-image, what are their social media practices and experiences with body positivity?”

My suggestion would be to reframe the aims into one that integrates the others and then rework your data analysis, either triangulating the results from FG and questionnaires or focusing in further analyzing one of them (probably the FG), making sure you accomplish all the stages of the thematic analysis.

One last observation: the title needs to be revised. Although the current one is catchy, it does not reflect the main message of your findings.

I hope these suggestions are helpful.

**Do you want your identity to be public for this peer review?** For information about this choice, including consent withdrawal, please see our Privacy Policy

Reviewer #1: No

Reviewer #2: **Yes: ** Dulce FERRAZ

---

## [Author Response · Author response to Decision Letter 1]

20 Sep 2024

Dear editor and reviewers.

Dear Dr. Annesha Sil

Thank you for considering our manuscript “How social media influences the self-image and body image among female and male adolescents – a qualitative study” for publication in PLOS ONE.

Thank you for your valuable and thorough feedback on our manuscript. We have included responses to all comments below, highlighted in track changes within the manuscript, and included the line number below where changes can be found. We have reworked the aims by incorporating the two methods, reworking the analysis, and triangulating the results from the focus groups and survey. Further, we have refined the results and removed some of the statements from the manuscript to keep focus on the main findings. Additional statements have been included as supporting information, along with the anonymous survey data. We hope our responses are satisfactory and that the manuscript meets the standards for publishing in PLOS ONE.

Yours sincerely

Merete Kolberg Tennfjord

Associate professor

Physiotherapist

Editor comments to the author

3. In this instance it seems there may be acceptable restrictions in place that prevent the public sharing of your minimal data. However, in line with our goal of ensuring long-term data availability to all interested researchers, PLOS’ Data Policy states that authors cannot be the sole named individuals responsible for ensuring data access (http://journals.plos.org/plosone/s/data-availability#loc-acceptable-data-sharing-methods).

Before we proceed with your manuscript, please also provide non-author contact information (phone/email/hyperlink) for a data access committee, ethics committee, or other institutional body to which data requests may be sent. If no institutional body is available to respond to requests for your minimal data, please consider if there any institutional representatives who did not collaborate in the study, and are not listed as authors on the manuscript, who would be able to hold the data and respond to external requests for data access? If so, please provide their contact information (i.e., email address). Please also provide details on how you will ensure persistent or long-term data storage and availability. Response: Due to participant privacy, the qualitative data material from the focus groups will not be shared. The online data is anonymous and will be shared as a supplementary file alongside this revision. We have updated this statement in the online submission system.

4. We note that you have included the phrase “data not shown” in your manuscript. Unfortunately, this does not meet our data sharing requirements. PLOS does not permit references to inaccessible data. We require that authors provide all relevant data within the paper, Supporting Information files, or in an acceptable, public repository. Please add a citation to support this phrase or upload the data that corresponds with these findings to a stable repository (such as Figshare or Dryad) and provide and URLs, DOIs, or accession numbers that may be used to access these data. Or, if the data are not a core part of the research being presented in your study, we ask that you remove the phrase that refers to these data. Response: Thank-you for this comment. We have removed the data related to age as these are not part of the research question.

5. Review Comments to the Author

Reviewer #1: 1. Is the manuscript technically sound, and do the data support the conclusions?

The manuscript is overall good. It presents a compelling exploration of an important topic in adolescent psychology, focusing on both females and males’ ideas and perceptions related to social media. The authors attempt to integrate various sides of the story when it comes to body image and self-image and try to touch on both. The results are interesting but maybe this section could benefit from further refinement in terms of organization and clarity. The research questions and data do support the conclusions in this manuscript.

Reply: Thank-you for your comment. We have revised the analysis and organized the results into two main themes and two sub-themes covering both the survey and the focus-group interviews. We have further removed some statements from the manuscript and included them as supplementary information organized under the different themes.

*2. Has the statistical analysis been performed appropriately and rigorously?

The statistical analysis of this manuscript appears to be appropriately and rigorously performed. The authors showed attention to detail and provided a strong foundation for the analysis. Response: thank-you for this comment.

*3. Have the authors made all data underlying the findings in their manuscript fully available?

No, the authors declared that there are some restrictions for data availability as well as that they have no consent to share data sampled from the interviews undertaken. Reply: Due to participant privacy, the qualitative data material from the focus groups will not be shared. The online data is anonymous and will be shared as a supplementary file alongside this revision. We have updated this statement in the online submission system.

*4. Is the manuscript presented in an intelligible fashion and written in standard English?

Overall, the manuscript presented in an intelligible fashion and written in standard English. There are a few minor grammar mistakes that are not that much important, but should be addressed with a brief review. Reply: We have reviewed the manuscript for grammatical errors and hope this work is satisfying.

*5. Review Comments to the Author

Notes:

- Abstract: missing the background of the topic. Response: Thank-you for your comment. We have included relevant background information for the abstract.

- Introduction: overall comprehensive and provides a well-established overview of the topic. Response: Thank-you for your positive feedback.

- Line 49-50: what is that sociocultural influence? Maybe give examples for the sentence is a bit ambiguous. Response: Thank-you for the comment. We have included some additional information for clarity and included the reference of Thompson et al 1999 that explains this relationship more in detail (line 58). We have also included some clarification related to personalization on social media and included a reference from our research group showing adolescents limited awareness of social media personalization (line 65).

- Limitation for the methodology section: sampling bias as the adolescents were recruited from one municipality. Response: thank-you for your comment. We agree that this information should be included, and we have included information about sampling bias (line 1127) and the cross-sectional design (line 1139) in the limitations. Related to the social desirability bias, we discussed this in the results section and further in the discussion, and also included a comment in the limitations section on line 1081.

- I do not know about using the adolescents first names in the results? Is that okay? Response: The adolescents from the interviews were given fictitious names to ensure anonymity. This information was included in line 360.

- The themes identified are very interesting, however they are presented in an excessively lengthy manner and could be rephrased to be shorter and more concise. Response: Thank-you for this comment. We have reworked the themes into two main themes and two sub-themes, triangulating the results from both the survey and the focus-groups.

- Discussion: well-structured and comprehensive discussion – compares and backs up the results of the study with previous literature, included both males and females. Also, there were different theory integration when explaining how social media effects self-image and body image, such as the social comparison theory. Response: Thank-you for your positive feedback. We have added a few points to the discussion following the reorganization of the results.

- The implications for clinical practice section in the discussion was important to add as it enhances the comprehensiveness and relevance of the study’s findings, giving an idea for targeted interventions and support services to better understand adolescents’ experiences with regards to social media and body image. Response: Thank-you for your positive feedback.

Reviewer #2: Thank you for the opportunity to review this manuscript.

The subject of self-image and its relations to social media is certainly relevant and there are many nuances to be explored within it. However, I think the manuscript has methodological issues that need to be addressed before it can be published.

1. Firstly, I lacks focus, trying to cover too many objectives and using different types of data that are not triangulated at the analytical and interpretative stages. The paper has currently four objetives: to investigate social media influences on adolescents` body-image; how adolescents view their body image in relation to their self image with emphasis on gender differences; to investigate adolescents’ social media practices; and their experiences with body positivity. My impression is that the thematic analysis was not fully completed, not reaching the stage when sub-themes are systematized into themes.

Response: Thank-you for your valuable comment. We agree that the objectives are too extensive, and we have rephrased accordingly (line 249). A detailed explanation of the re-organisation of the methods, analysis and results, triangulating the two datasets are discussed below under point 3.

2. Secondly, the data collection procedures have limitations that need to be considered in your analysis. The main one refers to the questions that were asked during FG, which induce participants to confirm that SM influences them and their body-image (“How does what you see on social media influence you?”) and some are not open questions (“Do you believe that social media contribute to body appearance pressure?”). The questions regarding SM use do not seem to have been fully analyzed or at least the results are not reported. Two other limitations are: 1. the fact that the FG were done in schools, an institutional scenario where adolescents will not necessarily feel comfortable to talk about themselves and their feelings, which may have contributed to their discourse being more influenced by social desirability; 2. only female researchers conducted the FG, which may have had an important impact on how boys expressed themselves (or refused to).

Reply: Thank-you for your comments. Related to the question: “Do you believe that social media contribute to body appearance pressure?”, this question gave us a more confirmatory response as the issues of body image pressure had already been explored through other questions. We have included some additional information on line 347. From this question, the adolescents also responded more objectively in relation to social media platforms as covered under theme 1.1. We have included more reflections on the questions asked in the limitations section.

Related to the question: “How does what you see on social media influence you?”, we have rephrased the question as it was posed: “In what way would you say that social media influence you?”. We acknowledge that this question may be leading in a way that the adolescents did confirm that SoMe influenced them. Based on the literature, a predominantly negative influence of SoMe has been published. Thus, we were interested in a more nuanced exploration covering negative and positive feelings towards this influence. We have included some more reflections in the limitations section related to the questions asked in both the survey and the focus-groups. However, we believe we have gained insight into various feelings and experiences of both self-image and body image among both males and females participating.

The questions “Is there any common theme on the social media accounts you follow?”, “Which social media accounts do you follow?”, and “What do you look at while on social media?” were included for the overall project and included in the interview but these questions gave no relevant data to the objectives of this study. We have therefore removed these questions from the current study.

Regarding the school as location, this was done for convenience due to the students taking time off from classes to join the project. Further, sampling students outside of the school environment could have resulted in groups of students not being familiar with each other, reducing the feeling of trust and safety among each other. We acknowledge that the school environment may for some be difficult and uncomfortable, but by creating various environments (same-gender and mixed-gender groups) we aimed to explore the social contexts within these different groups. In line 300 and 325 we have elaborated on the planning of the focus groups and included some information about the choices made for location.

Regarding the inclusion of a female interviewer, we agree that including a male interviewer could have made the male adolescents speaking more openly about social media influence. We have included this information in the limitations section.

3. When we get to the analysis, the results are presented in the form of two separated research questions, reflecting the lack ofdata triangulation. In addition, the first research question contains three questions in ti: "How do social media influence adolescents’ body-image, what are their social media practices and experiences with body positivity?” My suggestion would be to reframe the aims into one that integrates the others and then rework your data analysis, either triangulating the results from FG and questionnaires or focusing in further analyzing one of them (probably the FG), making sure you accomplish all the stages of the thematic analysis.

Reply: We have rephrased the aims into one that incorporates both datasets. We have further included a separate section explaining the triangulation of the analysis for the survey and the focus-groups (line 456). The results are thus presented including both data in an integrated manner covering adolescents’ feelings of their self-images and body images. We have refined the limitations section accordingly.

One last observation: the title needs to be revised. Although the current one is catchy, it does not reflect the main message of your findings. Reply: Thank-you for your comment. We have changed the title accordingly.

---

## [Decision Letter · Decision Letter 1]

14 Feb 2025

Dear Dr. Tennfjord,

Thank you for submitting your manuscript to PLOS ONE. After careful consideration, we feel that it has merit but does not fully meet PLOS ONE’s publication criteria as it currently stands. Therefore, we invite you to submit a revised version of the manuscript that addresses the points raised during the review process.

Reflect on the additional comments made by Reviewer 2, who is re-reviewing the paper based on changes. Consider Reviewer 3's (new reviewers) comments, which overlap in areas with Reviewer 2. Please ensure that the literature in this field is adequately covered as suggested by Reviewer 3Any study design comments should be fully addressed so the readership can understand the study design and analysis in relation to wider results in full.

We look forward to receiving your revised manuscript.

Kind regards,

Emily Lowthian

Academic Editor

PLOS ONE

Additional Editor Comments:

We were unable to source one of the original reviewers, so we have had to find a new reviewer. Please see the additional changes from Reviewer 2 - the original reviewer. In addition, Reviewer 3 - the new reviewer - has some additional comments which are of a similar nature to Reviewer 2 which will improve the paper.

Reviewers' comments:

Reviewer's Responses to Questions

**Comments to the Author**

Reviewer #2: All comments have been addressed

Reviewer #3: (No Response)

2. Is the manuscript technically sound, and do the data support the conclusions?

Reviewer #2: Yes

Reviewer #3: Yes

3. Has the statistical analysis been performed appropriately and rigorously?

Reviewer #2: I Don't Know

Reviewer #3: Yes

4. Have the authors made all data underlying the findings in their manuscript fully available?

Reviewer #2: No

Reviewer #3: Yes

5. Is the manuscript presented in an intelligible fashion and written in standard English?

Reviewer #2: Yes

Reviewer #3: Yes

Reviewer #2: Thank you for addressing the concerns I raised in the previous review. This version of the paper is clearer, but some sections still require further attention, as detailed below:

- The first sentence of the abstract ("The primary driver for social comparison is social media, where upward comparisons seem to harm young people's self-images and body images") is not fully correct and not in accordance with the first sentence of the introduction. I suggest replacing it with the latter.

- Several sentences could benefit from a more scientific tone, particularly by introducing some nuance or uncertainty. For instance, consider revising the use of words like "obvious" and "discovered" in the following examples:

- "It was obvious from our results that most adolescents viewed the impact of social media as negative."

- "Triangulating these data, we discovered the overall negative influence of social media on their self-image and body image.”

- In the triangulation topic, the following information that should be in the results, not methods : ”It was obvious from our results that most adolescents viewed the impact of social media as negative. Further, the descriptions of adolescents' self-images were related to appearance and unattainable body ideals. Thus, the findings from the survey aligned with the results of theme one, as identified from the focus group analysis. For theme two, the survey participants' positive responses aligned with the focus groups' responses, with some adolescents actively pursuing positive content on social media to improve their self-image and body image. However, a more in-depth understanding of this behavior, particularly among male participants, emerged from the focus groups. Lastly, experiences of body positivity were evident only from the focus groups and were not supported by survey data.”

- The categorization process in the survey is unclear. The sentence "Dividing the survey statements into categories, it was indicated that 55 respondents (44.4%) reported that social media had a negative influence on their self-image, compared with 20 (16.1%) who provided positive reports and 49 (39.5%) who reported mixed or neutral feelings (Table 2)" needs further clarification. Also, was the difference between negative and mixed-feeling responses statistically significant?

- The authors affirm that "Most of their described self-images were related to appearance and unattainable body ideals, giving the impression that body image plays a significant role in adolescents' self-images, more so for females than males. Pia, aged 16, said, "My self-image is influenced by ideals I know I cannot fulfill.” The quote does not explicitly refer to body image, and could be about other aspects of self-image, such as routine, housing, or leisure, which have also been shown to be influenced by social media in the literature.

- In several parts of the results and discussion, the concepts of body image and self-image seem to be conflated. For instance, in the conclusions, you state: "in the survey, we strived to capture the adolescents’ reflections on their self-image and how body image could relate. However, some may have been unfamiliar with the term self-image, which could reflect a relatively narrow comprehension of participants` self-image and, thus, their reflections on body”. How can you be certain that respondents were unfamiliar with the term "self-image" in an online survey?

- In the following sentence, there is no contradiction, but addition: Some participants mentioned in the survey that Instagram influenced their self-image. However, in the interviews, when asked whether social media contributes to body-image pressure, the participants suggested that this is most prevalent on Instagram, followed by Snapchat and TikTok

- In the opening sentence of the discussion ("This study has provided insight into the experiences of females and males with social media and how social media influences their self-image and body image”)- please add where (Norway)

- In the discussion, you evoke results regarding health that do not seem to have been presented in the manuscript and that exceed your objectives: "Goodyear et al. [35] found that health-related information, such as on diet and exercise, 770 can positively affect the health-related behaviors of young males and females. However, 24% of the younger adolescents in the latter study reported that the health-related content they see on social media had a negative impact on their health-related behaviors.”

- I do not see data supporting the age variation you refer to in this sentence:"Although these are positive findings for VSCO, this platform was only mentioned twice, both times by female participants, suggesting that the impacts of less approval-oriented platforms can vary with age and gender”.

- Considering that your data provides more substantial evidence of social media’s effects on females, I was surprised by the first conclusion: "greater attention must be paid to the influences of social media on males." Having read both versions of the manuscript, and noting that the first title focused on males, it seems that there may be a particular interest in studying this group, which could introduce a bias in your interpretation.

Reviewer #3: I commend the authors on their examination of the impact of social media on adolescents' self-image and body image. The combination of focus group interviews (qualitative) and survey data (quantitative) strengthens the study by providing in-depth insights. Many studies on body image focus on female experiences, but this paper includes both males and females, offering an important contribution to the literature on gender differences in how adolescents experience social media influence. While the study provides valuable findings, some areas require improvement:

Abstract

1. Additional demographic details such as the percentage of female participants in both studies and the mean age of participants should be included for better contextualization of the sample.

2. Additionally, a minor grammatical correction is needed in line 36: “Triangulating these data, discovered the overall negative influence of social media on their self-image and body image.’. It can be corrected to, “Triangulating these data revealed the overall negative influence of social media on their self-image and body image.”.

Introduction section

3. In lines 49-53, “Social media is considered the most pervasive and persuasive of these drivers. This is due to applications being largely visual platforms, which ceaselessly expose young people to a wide assortment of idealized images that reflect appearance and performance ideals as well as other personal characteristics.”, this claim needs a citation to support it, as it makes a broad assertion about social media’s role in sociocultural influence. For example, the authors can use data from Statista.

Dixon, S. (2023). Social media—Statistics & Facts. Statista. https://www.statista.com/topics/1164/social-networks/

4. Additionally, there is no dedicated paragraph on how social media influences self-image and body image. The authors should include a paragraph addressing current literature on social media’s associations with self-image and body image.

Given this, it should be noted that the authors should present a balanced viewpoint of both negative and null associations of social media on body image. For example, recent studies have found both negative (e.g., Fardouly & Vartanian, 2016) and null associations between objective social media screen time and body image dissatisfaction (Goh et al., 2025). There should be more literature review on both positive and negative findings of the associations.

Fardouly, J., & Vartanian, L. R. (2016). Social Media and Body Image Concerns: Current Research and Future Directions. Current Opinion in Psychology, 9, 1–5. https://doi.org/10.1016/j.copsyc.2015.09.005

Goh, A. Y. H., Hartanto, A., Kasturiratna, K. T. A. S., & Majeed, N. M. (2025). No Consistent Evidence for Between- and Within-Person Associations Between Objective Social Media Screen Time and Body Image Dissatisfaction: Insights From a Daily Diary Study. Social Media + Society, 11(1), 20563051251313855. https://doi.org/10.1177/20563051251313855

Methods; Focus Group Interviews and The Survey

5. In line 181 and line 266, there is clarification needed with regard to how screen time was measured (“The mean time spent on social media by the 48 participating adolescents was 4.8 hours per day, and was the same for both genders.” and “The time spent on social media was 4.48±2.30 hours overall, with females spending 4.74±2.24 hours and males spending 4.16±2.34 hours.”). Is this self-reported screen time, or was it objectively measured through an app or tracking software?

If self-reported, the authors should acknowledge the potential bias in recall accuracy.

Wilcockson, T. D. W., Ellis, D. A., & Shaw, H. (2018). Determining Typical Smartphone Usage: What Data Do We Need? Cyberpsychology, Behavior, and Social Networking, 21(6), 395–398. https://doi.org/10.1089/cyber.2017.0652

6. The authors should also list the 15 questionnaire items or provide more specificity regarding the aspects of social media measured, line 252, “The survey contained demographic questions that included gender, 15 questionnaire items on different aspects of social media.”.

7. Given that the survey had a low response rate (39.62%), it raises concerns about self-selection bias. The Discussion section should address this and discuss how it affects generalizability.

8. The authors categorized open-ended responses into positive, negative, and neutral, but do not specify how intercoder reliability was measured. Given this, agreement rates between authors, if applicable, can be reported to ensure transparency.

9. Was a thematic map used to organize qualitative findings? If so, including it as a figure would improve clarity.

Results and Discussion section

10. The results suggest that participants felt Instagram influenced their self-image the most, followed by Snapchat and TikTok (lines 358-361, “However, in the interviews, when asked whether social media contributes to body-image pressure, the participants suggested that this is most prevalent on Instagram, followed by Snapchat and TikTok.”). The authors should clarify whether this ranking is due to the nature of these platforms or simply because they are the most used applications.

If platform usage frequency explains this pattern, it should be highlighted in the discussion section as a limitation.

This manuscript offers valuable insights into the influence of social media on adolescents' self-image and body image, particularly in relation to gender differences. However, several areas require improvement. The introduction would benefit from a stronger theoretical foundation, particularly through the inclusion of relevant literature on social media’s associations with self-image and body image. In the methods section, clearer explanations are needed regarding screen time measurement, questionnaire items, and the potential impact of the survey’s low response rate on generalizability. Additionally, clarifications in data reporting—such as resolving inconsistencies in screen time estimates and providing transparency on intercoder reliability—are necessary to meet academic standards. Addressing these revisions will significantly enhance the manuscript’s clarity, methodological rigor, and overall impact.

**Do you want your identity to be public for this peer review?** For information about this choice, including consent withdrawal, please see our Privacy Policy

Reviewer #2: No

Reviewer #3: No

---

## [Author Response · Author response to Decision Letter 2]

12 Aug 2025

Response to reviewers have been uploaded as a separate document

---

## [Decision Letter · Decision Letter 2]

1 Oct 2025

How social media influences the self-image and body image among female and male adolescents

PONE-D-24-02516R2

Dear Dr. Tennfjord,

We’re pleased to inform you that your manuscript has been judged scientifically suitable for publication and will be formally accepted for publication once it meets all outstanding technical requirements.

Kind regards,

Emily Lowthian

Academic Editor

PLOS ONE

Reviewers' comments:

Reviewer's Responses to Questions

**Comments to the Author**

Reviewer #2: All comments have been addressed

Reviewer #3: All comments have been addressed

2. Is the manuscript technically sound, and do the data support the conclusions?

Reviewer #2: (No Response)

Reviewer #3: Yes

3. Has the statistical analysis been performed appropriately and rigorously?

Reviewer #2: (No Response)

Reviewer #3: Yes

4. Have the authors made all data underlying the findings in their manuscript fully available?

Reviewer #2: (No Response)

Reviewer #3: No

5. Is the manuscript presented in an intelligible fashion and written in standard English?

Reviewer #2: (No Response)

Reviewer #3: Yes

Reviewer #2: (No Response)

Reviewer #3: The authors have sufficiently addressed my comments given, and I appreciate their efforts. Thank you.

**Do you want your identity to be public for this peer review?** For information about this choice, including consent withdrawal, please see our Privacy Policy

Reviewer #2: **Yes: ** Dulce Ferraz

Reviewer #3: No

---

## [Editor Report · Acceptance letter]

PONE-D-24-02516R2

PLOS ONE

Dear Dr. Tennfjord,

I'm pleased to inform you that your manuscript has been deemed suitable for publication in PLOS ONE. Congratulations! Your manuscript is now being handed over to our production team.

Kind regards,

on behalf of

Dr. Emily Lowthian

Academic Editor

PLOS ONE